# Ontological Representation of Smart City Data: From Devices to Cities

**Paola Espinoza-Arias \***, **María Poveda-Villalón**, **Raúl García-Castro** and **Oscar Corcho**

Ontology Engineering Group, ETSI Informáticos, Universidad Politécnica de Madrid, 28660 Madrid, Spain;
mpoveda@fi.upm.es (M.P.-V.); rgarcia@fi.upm.es (R.G.-C.); ocorcho@fi.upm.es (O.C.)
**\*** Correspondence: pespinoza@fi.upm.es; Tel.: +34-913363670; Fax: +34-913524819

**Abstract:** Existing smart city ontologies allow representing different types of city-related data from cities. They have been developed according to different ontological commitments and hence do not share a minimum core model that would facilitate interoperability among smart city information systems. In this work, a survey has been carried out in order to study available smart city ontologies and to identify the domains they are representing. Taking into account the findings of the survey and a set of ontological requirements for smart city data, a list of ontology design patterns is proposed. These patterns aim to be easily replicated and provide a minimum set of core concepts in order to guide the development of smart city ontologies.

**Keywords:** ontology; smart cities; ontology design patterns

## 1. Introduction

The term smart city refers to a city that manages, in an intelligent way, all its associated resources with the aim to enhance the quality of the services provided to citizens and to improve their quality of life [1,2]. The smart city domain has been a topic of interest in many sectors around the world. Standardization bodies, for example, the International Telecommunications Union (https://www.itu.int) and the International Standards Organization (https://www.iso.org), have been working on defining standards and recommendations in order to provide a unified way to refer to and manage this particular field. In addition, several initiatives and projects in the smart city field have emerged, which denote the efforts and investments that industries, countries, and regions are making in order to manage the city resources in a better manner. In the case of city coalitions or research groups (e.g., Smart Cities-European Medium-Sized Cities (http://smart-cities.eu), Open and Agile Smart Cities (http://www.oascities.org), Smart Cities Council (http://smartcitiescouncil.com), etc.), their studies and business reports mention more than 300 smart cities involved [3] with an increasing need for and interest in exploring solutions in order to improve their city processes. In the case of projects (e.g., ESPRESSO [4], CityPulse [5], SmartSantander [6], etc.), which are financed by public, private, or a mix of both funds, they aim, in most cases, to provide technological tools to solve requirements in several city challenges.

In this respect, there is wide agreement about the fact that smart cities are characterized by a pervasive use of information and communication technologies [7–9], which, in various urban domains, may help cities make better use of their resources [10]. Some of these technologies include open data infrastructures, mobile applications, public participation tools, Internet of Things (IoT) platforms, etc. The data handled or produced by all these technologies is very heterogeneous in terms of formats, structure, and delivery mechanisms, both inside the same city and across different cities. Hence, this opens up the opportunity to create common models to allow interoperability inside cities. In this context, ontologies, understood as formal specifications of shared conceptualizations [11], can be used

to provide such models in a reusable and extensible manner. Several smart city ontologies have been developed in order to represent data related to cities (e.g., [12–16]). However, these ontologies rely on very specific ontological commitments for their particular needs, which makes data interoperability harder to achieve.

In order to minimize these strong commitments and to avoid developing large monolithic ontologies, small and generic ontology snippets, which capture key and reusable knowledge, may be used [17]. In ontological engineering these snippets have been used since the early 2000s and are known as Ontology Design Patterns (ODP). ODPs are inspired by software design patterns and can be understood as a reusable modeling solution to a recurrent ontology design problem [18]. Regarding the above-mentioned design strategy, a related approach for smart city knowledge management has been addressed in [19]. In this work, nine patterns have been identified based on the analysis of the Toronto 311 web pages. However, these patterns are only focused on representing municipal knowledge from citizen requests, and some of them rely on specializations that could be generalized (e.g., education, citizen, species, etc.). In addition, this work does not include important topics such as the measurements produced by several devices installed in the cities or indicators for tracking different aspects of city performance.

In this context, the use of ODPs for smart city models may facilitate ontology development. In addition, as explained in [20], taking into account that the use of ontology catalogs for the IoT and smart city domains is important in order to provide the developers a way to find, choose and reuse the ontologies that fit their needs, providing a list of ODPs to support core domain representation for smart city data may be also helpful. To the best of our knowledge, there is not a list of ODPs for the smart city domain, and the available ODPs in the general ODP catalogs (http://ontologydesignpatterns.org) do not cover all the usual ODPs that may be used in the context of the smart cities well.

In summary, the goal of this research is two-fold: carrying out a survey to provide a characterization of available smart city ontologies and providing a list of ontology design patterns for smart city domains based on the selected ontologies in the survey and on a set of general ontological requirements for smart city data.

This article is structured as follows. The methodological approach and results of our literature review referring to smart city ontologies are presented in Section 2. In Section 3, a characterization of the detected smart city ontologies is presented. Then, in Section 4, a list of ODP for smart cities is shown. Finally, some conclusions and future work are given in the last section.

## 2. Research Methodology

A three-step methodology has been followed in order to identify those ontology design patterns that are common in smart city ontologies, as shown graphically in Figure 1. The first step consists of a systematic literature review in order to identify the developed ontologies for smart cities, following the guidelines defined by Kitchenham and Charters [21]. The second step encompasses the characterization of the selected ontologies in the previous step, analyzing some associated ontology metadata and inspecting the ontology code in order to get specific details from them. Finally, the third step involves the analysis of such ontologies and the identification of common patterns based on ontological reverse engineering [22].

According to the aforementioned guidelines for conducting systematic literature reviews [21], the process usually contains three phases, namely planning, conducting, and reporting the review. The rest of this section is devoted to the detail of the planning and conducting phases in Sections 2.1 and 2.2, respectively. The reporting of the review is materialized in our work as the characterization of the selected ontologies, which will be described in Section 3, while the ontology design patterns' identification is detailed in Section 4.

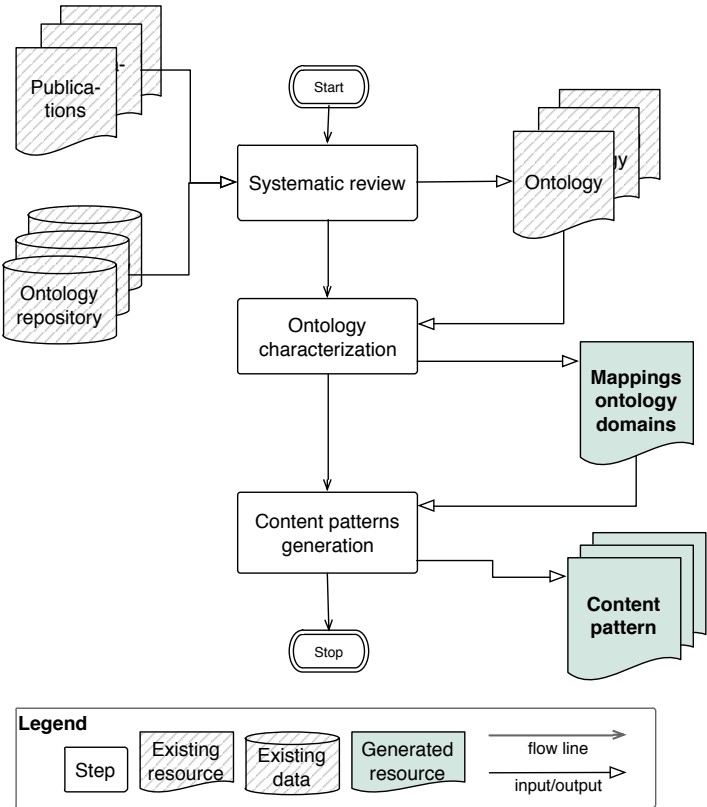

**Figure 1.** Three-step methodology to identify ontology design patterns in smart cities' ontologies.

## 2.1. Planning the Review

The main objective of this phase is to describe how the review protocol will be performed. To do so, the following points should be addressed: (a) the research question; (b) the source selection and search; (c) the inclusion and exclusion criteria; and (d) the selection procedure.

The main goal of the work being reported in this paper is to find those ontology design patterns that are used or appear in ontologies developed to support smart city use cases. Thus, we intend to answer the following research question: What are the most common ontology design patterns used in smart city ontologies?

In order to define the source selection and search, abstract and citation databases of peer-reviewed literature, specialized journals, and ontology indexes and catalogs have been taken into account. More precisely, Table 1 presents the selected sources for the search.

**Table 1.** Sources for the search.

| Source | Type | Entry Point |
|---|---|---|
| Scopus (SC) | database | https://www.scopus.com |
| Web of Science (WoS) | database | http://apps.webofknowledge.com |
| Semantic Web Journal: Interoperability, Usability, Applicability (SWJ) | journal | https://content.iospress.com/journals |
| Journal of Web Semantics: Science, Services and Agents on the World Wide Web (JWS) | journal | https://www.sciencedirect.com |
| Linked Open Vocabularies (LOV) | ontology index | https://lov.linkeddata.es |
| Linked Open Vocabularies for Internet of Things (LOV4IoT) | ontology catalog | http://lov4iot.appspot.com |
| smartcity.linkeddata.es | ontology catalog | http://smartcity.linkeddata.es |

Despite the review having included important databases, two specific journals have also been considered because they have published papers on the ontology topic. In the case of the ontology registries (indexes and catalogs), Linked Open Vocabularies (LOV) [23] was chosen because it contains general and domain ontologies, and Linked Open Vocabularies for Internet of Things (LOV4IoT) [24] and smartcity.linkeddata.es [25] were selected because they are ontology catalogs focused on smart cities. It is worth noting that LOV is classified as an ontology index as it has indexed the ontology OWL encoding, providing search features within the registered ontologies. LOV4IoT and smartcity.linkeddata.es are considered catalogs as they compile pointers to ontologies and classify them by domains, attaching tags, but searching within the ontology elements is not allowed.

Once the sources have been specified, some search terms were defined. These terms were set based on the research question and according to the main cross-domain topics involved in smart cities, oriented toward the Internet of Things domain as one of the main technologies deployed in those cities [26]. The search was limited to the following terms: ontology, ontologies, smart city, smart cities, pattern-based, design pattern, sensor, sensors, actuator, and actuators.

While carrying out the review, the particularities of each source were taken into account. In the case of databases and journals, connection words, OR and AND, were used in order to combine these terms in a search string, e.g., ("ontology" OR "ontologies") AND ("smart city" OR "smart cities"). The search strings were applied in the meta fields available in the electronic database entry points shown in Table 1. According to each source, the search strings were used in the fields: content, title, abstract, and keywords. Additionally, in the case of the databases' search, articles, books, and chapter books in English were considered.

In the case of ontology registries, as already mentioned, LOV is the only one that provides a search engine to find vocabulary terms, vocabularies, or agents matching keywords and/or filters; thus, the search terms were applied on vocabulary terms. Despite that LOV4IoT and smartcity.linkeddata.es do not provide a search engine, the search terms were applied over the domains used to classify the ontologies. In this cases, the terms "ontology" and "ontologies" were omitted from the search as they are implicit in these catalogs.

In order to detect those primary studies that provide valuable information to the research question, the inclusion and exclusion criteria were defined, as shown in Table 2.

**Table 2.** Inclusion and exclusion criteria.

| Inclusion Criteria (IC) | Databases | Journals | LOV | LOV4IoT | Smartcity. linkeddata.es |
|---|:---:|:---:|:---:|:---:|:---:|
| Peer-reviewed studies | ✓ | ✓ | | | |
| Studies published between 2012 and 2018 | ✓ | ✓ | | ✓ | |
| Studies that presented ontologies for smart city initiatives | ✓ | ✓ | ✓ | ✓ | ✓ |
| Ontology design patterns papers | ✓ | ✓ | | ✓ | |
| Ontology engineering papers | ✓ | ✓ | | ✓ | |
| **Exclusion Criteria (EC)** | | | | | |
| Ontologies whose code has not been published online | ✓ | ✓ | ✓ | ✓ | ✓ |
| Ontologies released before 2012 | ✓ | ✓ | ✓ | ✓ | ✓ |
| Studies/ontologies not written in English | ✓ | ✓ | ✓ | ✓ | ✓ |

After performing the search, all duplicate results were removed, and only one of them was kept in order to be analyzed. Taking this into account, in order to carry out the selection procedure the inclusion criteria and exclusion criteria were applied to these results.

## 2.2. Conducting the Review

The review process was performed by one of the authors and validated by the rest of the authors (the findings of this review process are available at https://delicias.dia.fi.upm.es/nextcloud/index.php/s/Nc8gxJAQFm6X7RM). Figure 2 shows the workflow of the review process. In this figure, numbers followed by letter "p" refer to publication results from journals and databases, and those followed by letter "r" refer to ontology registries' (catalogs and indexes) results. On the one hand, using the search strings, 119 studies were retrieved from the databases and specific journals. Then, 34 studies were discarded because they did not include an ontology. Next, 46 duplicate studies were removed, and 49 studies remained. After, 35 studies were excluded because they did not relate to the IoT domain, and finally, 2 studies remained, which included 2 ontologies.

On the other hand, 49 results were retrieved from the ontology registries. Then, 13 duplicate ontologies were eliminated, and 36 ontologies were kept. Next, 8 ontologies were excluded because they did not relate to the IoT domain. After, 8 ontologies were removed because they were not available online, and one ontology was also discarded because it was available only in Italian. Finally, 6 ontologies were excluded because they were not released after 2012, and 13 ontologies remained at the end of the review process. The results from both journals and ontology registries were merged, and the at end, 15 ontologies were obtained. These ontologies are listed in Table 3 as part of the ontology characterization section.

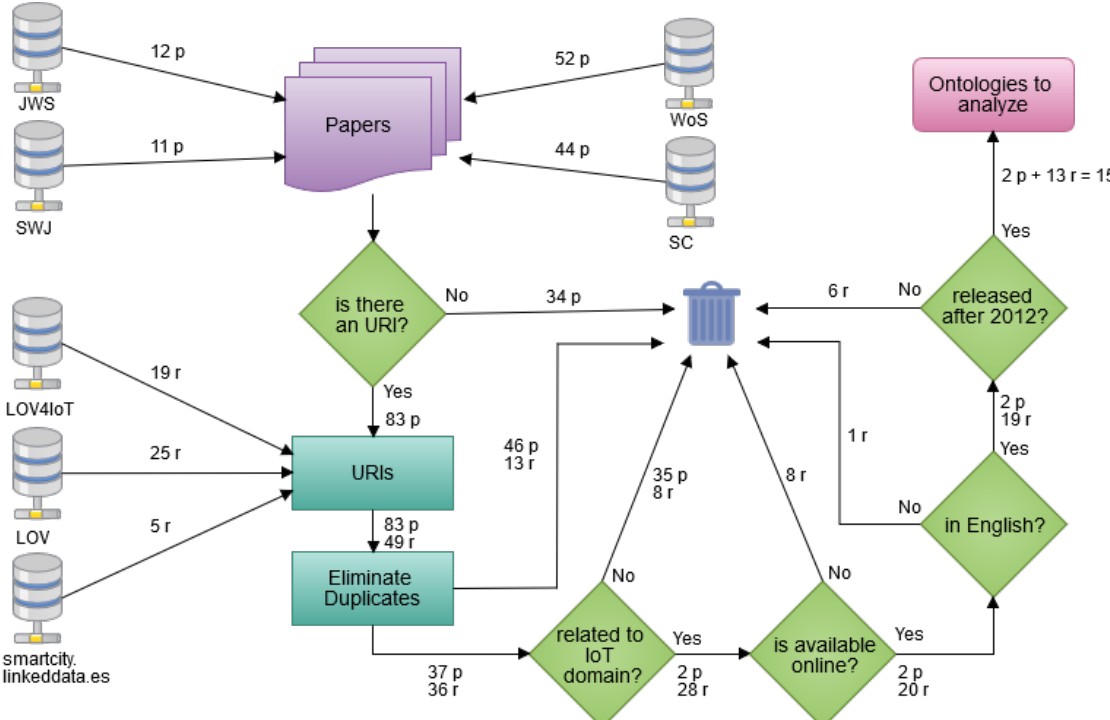

**Figure 2.** Systematic literature review workflow with the detail of the number of studies found. Numbers followed by letter "p" refer to publications results from databases and journals, and those followed by letter "r" refer to ontology registries' results.

It is worth mentioning that at the beginning of this search process, the Ontology Design Patterns' catalog was also considered as a potential source. This catalog collects a list of ODPs in a catalog classified by types of patterns. The patterns classified as Content Patterns (CPs) were checked in order to find those available for the smart city domain. As a result, the "Smart City Strategy Design" CP was found, but it has not been reviewed since it is used for the design of strategies to transform a city into a smart one, a topic that is out of the scope of this paper. In addition, this catalog does not have a domain for sensors or actuators; thus, it was excluded from the review.

### 3. Ontology Characterization

A characterization of the 15 ontologies obtained during the review selection was carried out. This characterization provides an overview of the gathered ontologies in order to get an initial idea about them and is two-fold: first, some general features (metadata) about the ontologies related to the development process or its availability were extracted; second, an examination of the domains included in the ontologies was performed, considering this time the "content" of the ontologies.

In the case of the metadata-oriented characterization, the following criteria were observed for each ontology:

- prefix: The prefix is a shortcut mechanism to refer to the ontology URI. As the scope of prefixes in RDF and semantic web technologies is local, that is different data or ontology providers may use different prefixes for a given ontology URI or namespace, in the context of this paper, the prefix chosen for each ontology has been taken from the ontology code. Otherwise, if the prefix was not available, a new one was defined by the authors or using prefix.cc (http://prefix.cc).
- name: The official name of the ontology.
- URI: The URI has been taken from the ontology code. If the ontology URI is not available online, a URL to the ontology code is provided.
- domain(s): A list of domains represented by the ontology according to the information provided in the journals or ontology registries.
- methodology: The name of the methodology followed for ontology development, if documented.
- standard: Identification of the standard if the ontology represents or is based on any.
- group/project: Name of the related project or group that developed the ontology.

The result of this characterization is shown in Table 3. In order to avoid confusion, the SSN ontology [27], originally denoted by the prefix *ssn*, has been assigned the prefix *oldssn*, while the prefix *ssn* is used for the module SSN of the new version [28]. In addition, the methodology and standard criteria were also omitted because few ontologies described them in their papers. For the methodology criterion, only in the *fiesta-iot* ontology paper was it mentioned that it was developed using the Ontology Development 101 methodology [29]. For the standard criterion, only the *gci* ontology clarified that the purpose was to represent indicators for city services and quality of life from ISO 37120 (https://www.iso.org/standard/62436.html).

As already mentioned, the second goal of the characterization is to analyze the ontology implementation in order to detect the domains represented and the reuse of other ontologies.

The first step consisted of the inspection of the ontology code in order to detect the reused ontologies by means of an import method (the use of *owl:imports* could be seen as an example of "hard reuse" as it entails strong ontological commitments) or by reference (the use of URIs identifying elements defined in other ontologies is considered "soft reuse"). The results obtained in this step are presented in Table 4. The *pep* and *sosa* [30] ontologies were omitted from this table because they are imported by *seas* and *ssn*; thus, they are represented as *seas/pep* and *ssn/sosa*, respectively.

**Table 3.** Ontology characterization.

| Prefix | Name | Reference | URI | Domain | Group/Project |
|---|---|---|---|---|---|
| fiesta-iot | FIESTA-IoT | Agarwal R. et al. [31] | http://purl.org/iot/ontology/fiesta-iot# | IoT | EU H2020 FIESTA-IoT Project |
| gci | Global City Indicator Foundation Ontology | Fox M.S. [32] | http://ontology.eil.utoronto.ca/GCI/Foundation/GCI-Foundation.owl | city performance indicators | Department of Mechanical and Industrial Engineering, University of Toronto |
| km4c | Km4city, the DISITKnowledge Model for City and Mobility | Bellini P. et al. [33] | http://www.disit.org/km4city/schema# | weather, cultural heritage, smart sensors, public structures, city parking, time, events, services transportation, geographic locations | SiiMobility Project |
| oldssn | The W3C Semantic Sensor Network Ontology | Compton M. et al. [27] | http://purl.oclc.org/NET/ssnx/ssn# | sensors, observations, sensor network | The W3C Semantic Sensor Network Incubator Group |
| pep | Procedure Execution Ontology | Lefrançois M. et al. [34] | https://w3id.org/pep/ | sensors, actuators | ITEA2Smart Energy-Aware Systems |
| sao | Stream Annotation Ontology | Puiu D. et al. [5] | http://iot.ee.surrey.ac.uk/citypulse/ontologies/sao/saov06.rdf | IoT, stream data, sensors, transport | CityPulse Project |
| san | Semantic Actuator Network | Seydoux N. et al. [35] | http://www.irit.fr/recherches/MELODI/ontologies/SAN# | actuators | IRIT and LAAS-CNRS |
| saref | SAREF: the Smart Appliances REFerence Ontology | Daniele L. et al. [36] | https://w3id.org/saref# | smart appliances, IoT, sensors, actuators, devices | European Commission and TNO |
| sco | Sensor Cloud Ontology | Müller H. et al. [37] | http://www.sense-t.csiro.au/sensorcloud/ontology# | sensors, IoT | Intelligent Sensing and Systems Laboratory, CSIRO |
| sctc * | STAR-CITY | Lécué F. et al. [38] | https://www.dropbox.com/s/xkvlb7a1fnibo4w/STAR-CITY-Ontologies.zip | sensors, IoT | STAR-CITY Project |
| seas | SEAS | Lefrançois M. et al. [34] | https://w3id.org/seas/ | IoT | ITEA2 Smart Energy-Aware Systems |
| smart-city | Smart City Ontology | Komninos N. et al. [39] | https://www.dropbox.com/s/q7tz39jjeeibhzl/2015-SMART%20CITY%20ONTOLOGY-V01.owl?dl=0 | smart applications | Department of Urban and Regional Development and Planning, Aristotle University of Thessaloniki |
| sosa | Sensor, Observation, Sample, and Actuator | Janowicz K. et al. [30] | http://www.w3.org/ns/sosa/ | sensors, actuators, observations, samples | The W3C/OGC Spatial Data on the Web Working Group |
| ssn | Semantic Sensor Network Ontology | Haller A. et al. [28] | http://www.w3.org/ns/ssn/ | sensors, actuators, observations, samples, sensor network | The W3C/OGC Spatial Data on the Web Working Group |
| vital | VITAL | Kazmi A. et al. [40] | http://vital-iot.eu/ontology/ns/ontology.owl | sensors, smart cities, sensors measurements, IoT | VITALProject |

* Prefix defined by the authors.

The results provided in this table suggest that the most reused ontologies are those stable recommendations and those published by standardization bodies like the W3C Semantic Sensor Network Ontology (*oldssn*) [27] (https://www.w3.org/2005/Incubator/ssn/ssnx/ssn), OWL-Time [41] (http://www.w3.org/2006/time), and geo [42] (https://www.w3.org/2003/01/geo/wgs84_pos#). In addition, by looking at the domains of the ontologies commonly reused, it is possible to infer which are the domains usually needed in smart city use cases. The inferred domains and their related ontologies are listed as follows:

- time: OWL-Time, timezone (http://www.w3.org/2006/timezone), and timeline (http://purl.org/NET/c4dm/timeline.owl).
- geospatial: geo, geosparql [43,44] (http://www.opengis.net/ont/geosparql), and geonames (http://www.geonames.org/ontology).
- observations and measurements: QUDT, qu, muo, ucum, OM [45], oldssn, and om.
- metadata: VANN (http://purl.org/vocab/vann), dc [46] (http://purl.org/dc/elements/1.1), dcterms (http://purl.org/dc/terms), etc.

In addition, other ontologies are reused to model very specific domains, for example goodrelations (http://purl.org/goodrelations/v1) and FOAF (http://xmlns.com/foaf/0.1). Even though the metadata, upper level ontologies, and specific domains as persons and e-commerce do appear in the smart city ontologies analyzed, such domains are not considered as smart city core domains in the scope of this paper, since our focus is on these domains directly related to the Internet of Things. A similar work where a rank with the most relevant concepts extracted from the IoT ontologies is presented is [47]. These concepts also allows us to infer the same domains detected in this work.

**Table 4.** Ontologies reused (ref = referenced, imp = imported, rm = referenced as metadata) by smart city ontologies.

| Analyzed \ Reused | OWL-Time | Timeline | Timezone | Geo | Geosparql | Geonames | QUDT | qu | muo | ucum | OM | oldssn | om | Goodrelations | vaem | dc | VANN | SKOS | Dcterms | FOAF | prov | Schema | DUL |
|---|---|---|---|---|---|---|---|---|---|---|---|---|---|---|---|---|---|---|---|---|---|---|---|
| fiesta-iot | ref | | | ref | | | | ref | | | | ref | | | | rm | rm | | rm | | | | ref |
| gci | imp | | | | | ref | | | | | ref | | | | | ref | rm | | | | ref | ref | |
| km4c | ref | | | ref | ref | | | | | | | | | ref | | rm | rm | | rm | ref | | ref | |
| oldssn | | | | | | | | | | | | | | | | rm | | | rm | | | imp | |
| sao | | ref | | | | | | ref | | | | ref | | | | rm | | | | | ref | ref | |
| san | ref | | | | | | ref | | | | | ref | | | ref | rm | rm | | rm | rm | | | |
| saref | imp | | | | | | | | | | | | | | | | | | rm | | | | |
| sco | | | | | imp | | | | ref | ref | | imp | imp | | | imp | | | | | | | |
| sctc | imp | | ref | imp | | | | | | | | | | | | rm | | | | | rm | | |
| seas/pep | imp | | | | | | | | | | | | | imp | | | rm | | rm | | | | |
| smart-city | | | | | | | | | | | | | | | | | | | | | | | |
| ssn/sosa | | | | | | | | | | | | | | | | rm | rm | | rm | | | rm | |
| vital | ref | | | ref | | | ref | | | | | ref | | | | | | | | | ref | ref | |

The second step of the characterization consisted of a further inspection of the ontologies' code in order to check the coverage of the domains for smart city ontologies in the IoT landscape. The results

of such inspection are shown in Table 5. The ontology code inspection was carried out in order to find classes, object properties, or data type properties related to each domain. For example, the event domain was detected in the *km4c* ontology by means of the class `km4city:Event`. This class has several data type properties such as `km4city:eventCategory`, `km4city:eventTime`, `km4city:placeName`, etc., which allows representing the type of event, the starting time of the event, the location where the event takes place, etc. (the findings of the ontology code inspection searching for ontology elements related to the observed domains are available at https://delicias.dia.fi.upm.es/nextcloud/index.php/ s/F2aFjEqSn6kKJZy).

**Table 5.** Domains represented in smart city ontologies.

| Domains / Ontologies | Administrative Area | City Object | Event | KPI | Public Service | Topology | Observations/ Measurements |
|---|---|---|---|---|---|---|---|
| fiesta-iot | | ✓ | | | | ✓ | ✓ |
| gci | ✓ | | | ✓ | ✓ | ✓ | ✓ |
| km4c | ✓ | ✓ | ✓ | | ✓ | ✓ | ✓ |
| oldssn | | ✓ | | | | | ✓ |
| sao | | ✓ | ✓ | | | | ✓ |
| san | | ✓ | | | | | ✓ |
| saref | | ✓ | | | | | ✓ |
| sco | | ✓ | | | | ✓ | ✓ |
| sctc | | ✓ | | | ✓ | ✓ | ✓ |
| seas/pep | | ✓ | | | | | |
| smart-city | ✓ | ✓ | ✓ | ✓ | ✓ | | |
| ssn/sosa | | ✓ | | | | | ✓ |
| vital | | ✓ | | | ✓ | ✓ | ✓ |

Below, a brief explanation about each domain represented in smart city ontologies is shown. The *Administrative Area* domain represents places delineated for jurisdiction purposes of a particular government (e.g., city, district, neighborhood, etc.). The *City Object* domain represents all objects that can be contained by a city (e.g., devices, buildings, transport means, etc.). The *Topology* domain represents all things that can have a spatial extension (e.g., roads, train stations, commercial premises, etc.). The *Event* domain describes activities performed in a city during a specified period of time (e.g., concerts, exhibitions, races, etc.). The *Key Performance Indicator* (KPI) domain represents the measured values, according to a method, in order to monitor the performance of a city (e.g., noise pollution level, air quality index, recycling rate, etc.). The *Public Service* domain involves all services provided by public administrations and organizations (e.g., waste management, public parking, water quality control, etc.). Finally, the *Observations and Measurements* domain represents all measured values related to a particular property of any feature of interest (e.g., noise levels, weather conditions, air quality, etc.).

## 4. Ontology Design Patterns Proposal for Smart City Ontologies

Table 5 shows that several domains are represented by the analyzed ontologies. However, it can be observed that no ontology covers all the core domains for smart cities. Given this situation, it is advisable to provide a common conceptualization that involves such domains in order to represent the data potentially involved in smart city use cases. As mentioned in the Introduction, this work is devoted to provide a list of domain-dependent patterns, specifically named ODPs.

In order to provide a set of ontological requirements for each core domain, an ontological reverse engineering has been performed. In this process, some smart city-related initiatives developed by standardization bodies, associations, and European projects have been taken into account. More precisely, the process is based on data models, ontologies, datasets, and standards provided by OGC (Open Geospatial Consortium http://www.opengeospatial.org),

FEMP (Spanish Federation of Municipalities and Provinces http://www.femp.es), ISO (International Organization for Standardization https://www.iso.org), FIWARE (A software platform for smart cities https://www.fiware.org), and AENOR (Spanish Association for Normalization https://www.aenor.com). Such a list of ontologies' requirements has been taken in the present survey as input in order to: (a) observe whether the analyzed ontologies model the given domains and to what extent and (b) propose a list of general ontological patterns to fulfill the proposed requirements.

Taking all this information and inputs into account, a list of ODPs for smart city ontologies has been developed in order to support the reuse and to ease the ontology design and conceptualization activities. With the purpose of describing each ODP, a unified template, adapted from [48], was designed. The template consists of the following fields:

- name: the given name for the pattern.
- description: a natural language explanation of the domain addressed by the pattern.
- requirements: the ontological requirements that the pattern addresses.
- graphical representation: a diagram to describe the structure of the pattern, depicted by means of UML-based notation, as explained in [49].
- use case: an example about how this pattern can be applied.
- note: remarks for clarifying some aspects of the pattern.

According to the ontological requirements, classes, and properties have been identified in order to model the ODPs. After that, graphical models (using UML-base notation) have been depicted and the properties have been split into object properties or data type properties according to the type of information they represent. Then, an iteration process has been performed in order to identify possible data annotation issues. Finally, the patterns have been validated by three of the authors. This validation included checking ontological requirements' coverage and checking alignment with existing ontologies and data models.

The defined ODPs are: *Administrative Area*, *City Object*, *Event*, *Key Performance Indicator* (KPI), *Measurements*, *Public Service*, and *Topology*. These patterns will be described in this order in the following subsections. However, due to the Topology ODP used in the description of the other patterns, it will be the first explained.

### 4.1. Topology ODP

This pattern has been inspired by the geospatial domain identified by analyzing the smart city ontologies in Section 3, which suggests two approaches. On the one hand, almost all the analyzed ontologies uses the *geo* ontology in order to represent the point where a certain element is located. On the other hand, only *km4c* uses *geosparql* in order to represent the geometry of certain concepts, for example a district, road, railway, etc. In order to design this pattern, the best practices provided by the W3C Spatial Data on the Web Best Practices Group (https://www.w3.org/TR/sdw-bp) have also been considered. These best practices present a comparison between common spatial data vocabularies, including the *geo* and *geosparql* ontologies, and rely on how to represent spatial data using them. In Table 6, the proposed Topology ODP is shown.

Taking into account the aforementioned best practices, this pattern follows the model provided by the *geosparql* ontology, since it allows us to represent spatial things, geometries, and several coordinate reference systems. More precisely, the *SpatialFeature* has been defined as a spatial thing that represents anything with spatial extent (https://www.w3.org/TR/sdw-bp/#dfn-spatial-thing). In addition, the *SpatialObject* has been included in this pattern in order to reuse the property *contains* from *geosparql*. Moreover, the *Geometry* concept has been used to represent the geometry, which defines a *SpatialFeature* (e.g., point, line, polygon, multi-line, etc.).

**Table 6.** Topology Ontology Design Pattern (ODP).

| Name | Topology |
|---|---|
| **Description** | |
| This pattern provides a model to represent anything that can have a spatial representation. | |
| **Requirements** | |
| **TOPO-1:**　　Spatial things are defined by a geometry. | |
| **TOPO-2:**　　A spatial thing can contain other spatial things. | |
| **TOPO-3:**　　A spatial thing can be located at a given point. | |
| **Graphical representation** | |

| **Use Case** |
|---|
| A city council usually provides a training service. The city council can have a registry with the geometry of all train stations. These train stations provide a parking service, which can contain parking slots defined by geometries. These slots could be supervised by some sensors located at a given point. The passengers could use a mobile application with the aim to obtain real-time information about what slot is available, and in this way, they could calculate the estimated time that they will spend between parking their car and boarding the train. |

### 4.2. Administrative Area ODP

This pattern has been detected in the *gci*, *km4c*, and *smart-city* ontologies. Each of these ontologies defines its own taxonomy in order to represent a city, municipality, district, etc. With the aim of defining the Administrative Area ODP, some requirements were extracted from the Territory Vocabulary (http://vocab.linkeddata.es/datosabiertos/def/sector-publico/territorio) included in the AENOR norm UNE 178301:2015 (The norm UNE 178301:2015, entitled "Smart Cities. Open Data", aims to provide a set of indicators to measure the maturity level of open data projects for smart cities; and it also provides a set of common vocabularies in order to ease the development and deployment of these open data projects. The norm is available at https://www.aenor.com/normas-y-libros/buscador-de-normas/une?c=N0054318). In addition, the GeoSPARQL standard was taken into account in order to represent the topology that an administrative area can have. Finally, various types of administrative units within the Member States of the European Union published by the Metadata Registry (http://publications.europa.eu/mdr/authority/atu-type/index.html) were also considered. Taking these requirements into account, the pattern shown in Table 7 is proposed for the *Administrative Area* domain.

A similar pattern for an administrative area is available in ISO/IEC 30182:2017 (https://www.iso.org/standard/53302.html), where there is a *Place* concept that represents a geographic or virtual part of space.

**Table 7.** Administrative Area ODP.

| Name | Administrative Area |
|---|---|

| **Description** |
|---|
| This pattern provides a model to represent a place delimited by the jurisdiction purposes of a particular government. |

| **Requirements** | |
|---|---|
| **ADM-1:** | A city is a large human settlement. A city is distinguished from other human settlements by its relatively great size, but also by its functions and its special symbolic status, which may be conferred by a central authority. |
| **ADM-2:** | A city can contain neighborhoods or districts. |
| **ADM-3:** | A neighborhood is a geographically-localized community within a larger city, town, suburb, or rural area. |
| **ADM-4:** | A district is a type of administrative division that, in some countries, is managed by local government. Across the world, areas known as "districts" vary greatly in size, spanning regions or counties, several municipalities, subdivisions of municipalities, school districts, or political districts. |
| **ADM-5:** | A district can belong to a city. |
| **ADM-6:** | An administrative division, unit, entity, area, or region, also referred to as a sub-national entity, constituent unit, or country subdivision, is a portion of a country or other region delineated for the purpose of administration. |
| **ADM-7:** | A neighborhood, district, city, etc., can contain a road. |
| **ADM-8:** | A city can be defined by a geometry. |
| **ADM-9:** | A district can be defined by a geometry. |
| **ADM-10:** | A country can be defined by a geometry. |
| **ADM-11:** | An Administrative Area can be defined by a geometry. |
| **ADM-12:** | A spatial thing can be an administrative area, which can be cities, districts or countries. |
| **ADM-13:** | A city can include different administrative areas. |
| **ADM-14:** | A spatial thing can belong to one or more administrative areas or other delimited areas like neighborhoods. |

| **Graphical representation** |
|---|

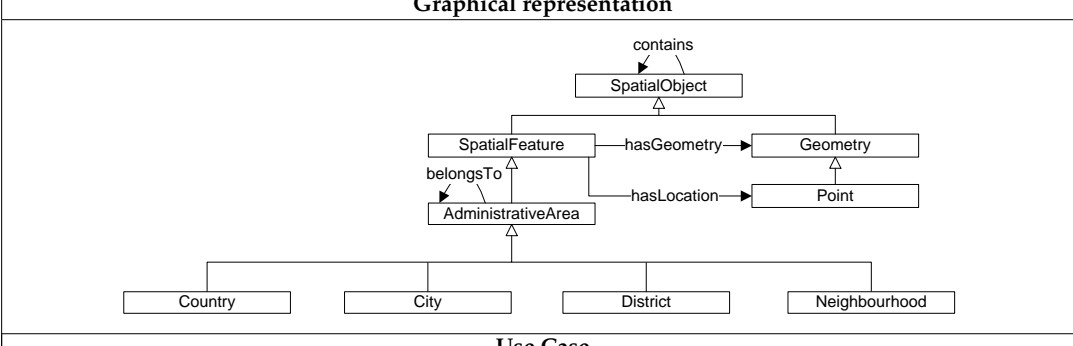

| **Use Case** |
|---|
| New York City is defined by a geometry. In order to manage its area, this city is organized into five districts, which are also defined by a geometry. Each district contains neighborhoods. Neighborhoods can contain roads, public infrastructures, and other city facilities. This area disposition is useful in order to organize their facilities. For example, each district is responsible for managing the complex and large road system that crosses it. In this context, several traffic restrictions have been adopted in order to control the traffic. By means of traffic cameras, each road collects traffic data, and these allow changing the patterns of traffic lights in real time. |

| **Note** | The requirements ADM -1, ADM-3, ADM-4, and ADM-6 have not been included in the diagram because they are part of the concept documentation. In addition, the requirement ADM-7 has not been explicitly shown in the diagram because the road may be represented by means of a subclass of the *SpatialFeature* concept according to the description provided in the Topology ODP. |
|---|---|

### 4.3. City Object ODP

As Table 5 shows, the *City Object* is the most represented domain by smart city ontologies. The most common city objects represented by these ontologies are devices, for example sensors. The requirements that this pattern addresses were extracted from the GeoSPARQL standard, geo vocabulary, and the CityGML (https://www.citygml.org/about) open standardized data model for digital 3D models of cities and landscapes.

In Table 8, the proposed City Object ODP is shown. The concept *CityObject* has been defined in order to represent the same class defined in the CityGML data model. In the CityGML model, several modules are defined in order to represent the class definitions for the most important types of objects within 3D city models; all of them are depicted as subclasses of the *CityObject*, as shown in the graphical representation from Table 8. The properties defined in CityGML for these models could be used to represent more specific details about each city object, for example their usage and function.

**Table 8.** City Object ODP.

| Name | City Object |
|---|---|
| **Description** | |
| This pattern provides a model for an object that belongs to a city. | |
| **Requirements** | |
| **OBJ-1:** A city object can be defined by a geometry. | |
| **OBJ-2:** A city object can be located in a given point. | |
| **OBJ-3:** A city can have city objects. | |
| **OBJ-4:** Each city object can have a location. | |
| **OBJ-5:** Some specific types of city objects can be: automated external defibrillator, urban furniture (bench, basket, street lights, etc.), cameras, or monuments. | |
| **OBJ-6:** A spatial thing can be a city object. | |
| **Graphical representation** | |

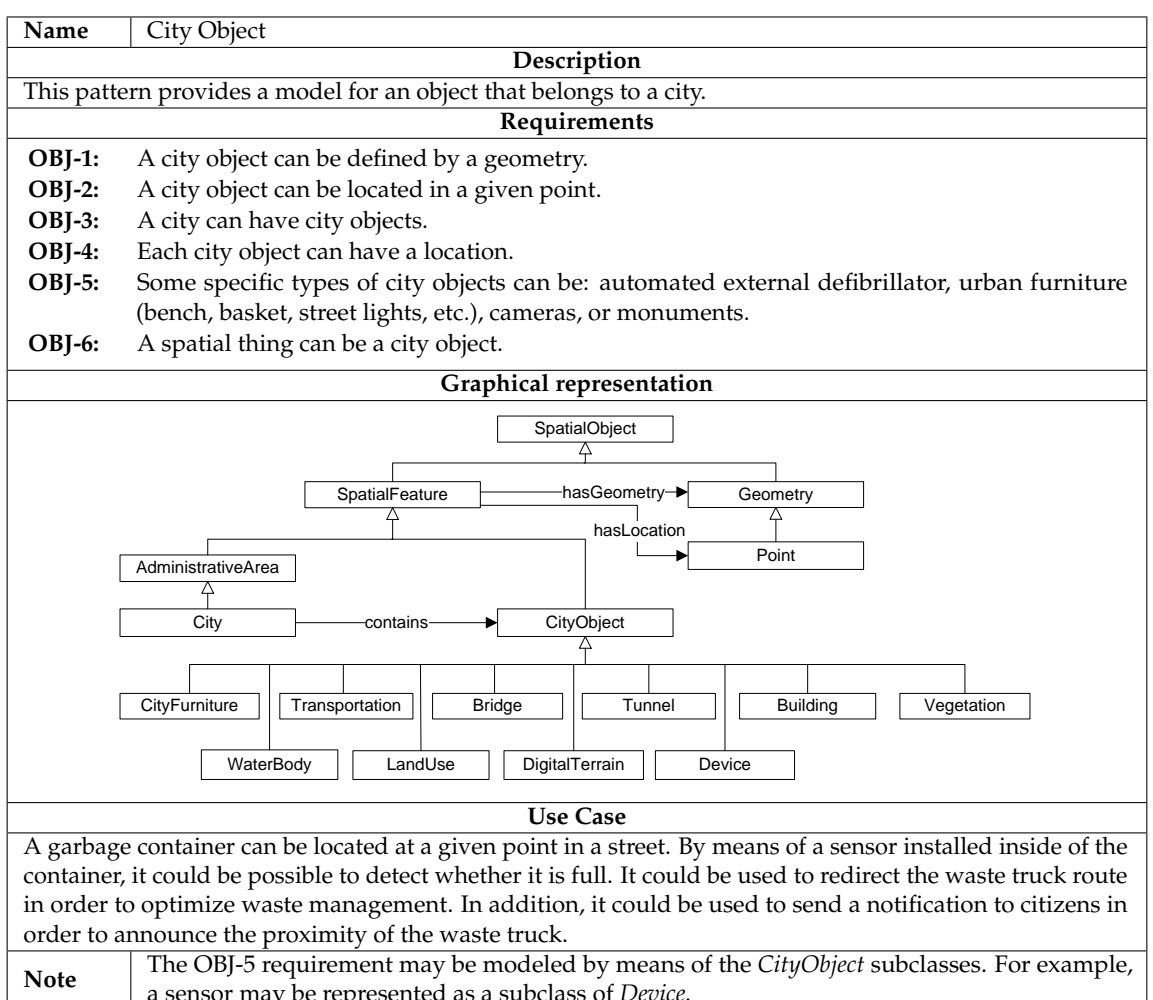

| **Use Case** | |
|---|---|
| A garbage container can be located at a given point in a street. By means of a sensor installed inside of the container, it could be possible to detect whether it is full. It could be used to redirect the waste truck route in order to optimize waste management. In addition, it could be used to send a notification to citizens in order to announce the proximity of the waste truck. | |
| **Note** | The OBJ-5 requirement may be modeled by means of the *CityObject* subclasses. For example, a sensor may be represented as a subclass of *Device*. |

### 4.4. Event ODP

The model for this pattern has been reengineered from two vocabularies. The first model is the schema.org vocabulary, and it is taken into account because the *km4c* ontology reuses this well-known model in order to represent an event. The second model is the Cultural Agenda Vocabulary (http://vocab.linkeddata.es/datosabiertos/def/cultura-ocio/agenda), referenced by the AENOR norm UNE 178301:2015, which is devoted to represent events in a city.

In Table 9, the proposed pattern for the *Event* domain is shown. The concepts *TemporalEntity*, *Interval*, and *Instant* have been represented in order to model the time requirements. These concepts have been taken from the OWL-Time ontology, which, according to Table 4, is the most reused ontology for time representation in smart city ontologies. The *hasBeginning* and *hasEnd* OWL-Time properties may be used in order to represent the starting/ending date/hour of the *Event*. In addition, the concept *Interval* has been extended by means of the class *NonConvexInterval*, which allows us to represent periodic intervals with gaps between them (e.g., "every Wednesday"), as is explained in [50].

**Table 9.** Event ODP.

| Name | Event |
|---|---|
| **Description** | |
| This pattern provides a model for something that occurs in a certain place during a particular interval of time. | |
| **Requirements** | |
| EVE-1: | An event is something that occurs in a certain place during a particular interval of time. |
| EVE-2: | An event can have a starting date. |
| EVE-3: | An event can have an ending date. |
| EVE-4: | An event can have a starting hour. |
| EVE-5: | An event can have an ending hour. |
| EVE-6: | An event can takes place in a given location. |
| EVE-7: | An event can have a given accessibility type. |
| EVE-8: | An event can be organized by one or more agents. |
| EVE-9: | There can be different types of events. |
| EVE-10: | An event can contain sub-events. |
| EVE-11: | There can be recurrent events. |

*Graphical representation*

*Use Case*

City councils usually organize public events during the summer. In general, these events could have free access and could be located in public spaces, for example a park. An event could be held at a certain date and time previously established. Depending on the kind of event, certain accessibility facilities should be specified, for example if the event is a concert, it will have reduced accessibility for deaf persons.

| Note | The requirement EVE-1 has not been included in the diagram because it is part of the documentation, for example it has been used in the description slot of this pattern. The requirement EVE-9 could be modeled in two ways. On the one hand, the concept *Event* could be extended in order to include subclasses that represent the different types of events. On the other hand, the concept *Event* could be linked to a specific concept from a controlled vocabulary of types of events. In this last approach, it will be necessary to include a property in order to link the concepts, and it could be named for example as *eventType*. Finally, the requirement EVE-11 could be represented by means of the concept *NonConvexInterval* linked to a concept from a controlled vocabulary where concepts such as daily, weekly, monthly, or annually could be defined in order to represent the recurrence of the *Event*. |
|---|---|

With the aim to avoid a particular commitment, the concept *Agent* has been defined. An *Agent* can be any individual, organization, or group that organizes an *Event*. The *Agent* could be represented by means of other ontologies, for example: FOAF, schema.org, etc. Finally, the property *accessibilityType* has been used to link the *Event* to an *Accessibility* concept, which represents a concept from a controlled vocabulary where the accessibility types could be previously defined.

It should be mentioned that in ISO/IEC 30182:2017, there is an *Event* concept described as an occurrence that has happened or might happen. Such a concept of *Event* could be considered as a generalization of the *Event* presented in the present ODP, which is more specific.

*4.5. Measurement ODP*

The Measurement ODP has been designed from the *Observations and Measurements* domain identified in Table 5. Together with the *City Object* domain, this domain is the most represented by smart city ontologies. With the aim of extracting the requirements that this pattern addresses, the *saref*,

*ssn-sosa*, *san*, and *fiesta-iot* ontologies have also been taken as input for the reverse engineering process. In Table 10, the proposed pattern for this domain is shown. In this pattern, the concept *Property* has been defined as anything that can be sensed, measured or controlled in a place. This concept may be extended in order to represent data if needed. In addition, the *UnitOfMeasure* concept has been defined in order to represent the units for the measurements. This concept could be modeled by means of well-known ontologies such as OM (http://www.wurvoc.org/vocabularies/om-1.8/), QUDT (http://qudt.org/schema/qudt), qu (http://purl.org/NET/ssnx/qu/qu), or those explained in [51].

Moreover, the *FeatureOfInterest* concept has been included in order to represent the thing whose property is measured. It is worth mentioning that this *FeatureOfInterest* is an abstraction of a real-world phenomena not only in a spatial sense, as the *SpatialFeature* concept defined in the Topology ODP. In this context, the *SpatialFeature* is represented as a subclass of the *FeatureOfInterest*. Finally, the *Device* concept, the same explained in the *City Object* domain, could be extended in order to represent a sensor, actuator, etc., that is measuring.

**Table 10.** Measurement ODP.

| Name | Measurement |
|---|---|
| **Description** | |
| This pattern provides a model for a method to represent the measured value of a property. | |
| **Requirements** | |
| **MEA-1:**   A measurement can be made by a device. | |
| **MEA-2:**   A device can be a sensor, actuator, etc. | |
| **MEA-3:**   A measurement can be related to a certain property. | |
| **MEA-4:**   A property can be light, pressure, height, etc. | |
| **MEA-5:**   A property can belong to a feature of interest. | |
| **MEA-6:**   A measurement can have a result value in a given timestamp. | |
| **MEA-7:**   A measurement can be measured in a certain unit of measurement. | |
| **Graphical representation** | |

| **Use Case** |
|---|
| A public administration building may provide a system that allows its employees to book meeting rooms. All meeting rooms have sensors to measure temperature, light, presence, and control the status of their available devices. On the basis of the initial time of the meeting, a sensor measures the light level; then, if necessary, the lights are turned on automatically. In addition, sensors start measuring the temperature a couple of minutes before the meeting and automatically turn on/off the air conditioner or the heating. During the meeting, the room automatically adjusts the temperature according to the values detected by the sensor in order to provide a pleasant place. When the meeting finishes, the employees leave the room, and the presence sensor does not detect any value; then, the lights and air conditioner/heating are turned off. |

Despite the Ontology Design Patterns catalog having been discarded as a search source, the inspection of the *san* ontology revealed that it is based on the actuation-actuator-effect pattern (http://ontologydesignpatterns.org/wiki/Submissions:Actuation-Actuator-Effect). This pattern, cataloged in the Internet of Things domain, intends to model the relationship between an actuator and the effect it has on its environment.

### 4.6. Key Performance Indicator ODP

This pattern has been inspired by the *gci* and *smart-city* ontologies. In addition, the models analyzed in order to define the requirements for this domain are the FIWARE data model for KPI (http://fiware-datamodels.readthedocs.io/en/latest/KeyPerformanceIndicator/doc/spec/index.html) and ISO:37120:2014 (https://www.iso.org/standard/62436.html). Table 11 presents the ODP for Key Performance Indicators. The *UnitOfMeasure* concept has been defined in order to represent the units in which the KPI is measured and may be modeled as was explained in the Measurement pattern. In addition, the concept *TemporalEntity*, as in the case of the Event ODP, has been included in order to represent the period of time over which the KPI is measured and the timestamp when the KPI value is calculated. Specifically, the OWL-Time concepts *Interval* and *Instant* could be used to represent both requirements.

**Table 11.** Key Performance Indicator ODP.

| Name: | Key Performance Indicator |
|---|---|
| **Description** | |
| This pattern provides a model for a method to measure the performance in a city in order to evaluate the success of an organization or of a particular activity in which it engages. | |
| **Requirements** | |
| **KPI-1:** | A Key Performance Indicator (KPI) is a type of performance measurement. KPIs evaluate the success of an organization or of a particular activity in which it engages. |
| **KPI-2:** | A KPI has a name. |
| **KPI-3:** | A KPI can assess an organization, service, process, or product. |
| **KPI-4:** | A KPI has a description. |
| **KPI-5:** | A KPI can be calculated by an organization. |
| **KPI-6:** | A KPI can aggregate different measurements. |
| **KPI-7:** | A KPI can have a calculation period over which it is measured. |
| **KPI-8:** | A KPI can take a given value in a given timestamp. |
| **KPI-9:** | A KPI can have a creation date. |
| **KPI-10:** | A KPI can have an expiration date. |
| **KPI-11:** | A KPI can have a last update date. |
| **KPI-12:** | A KPI might refer to a given location of an area. |
| **KPI-13:** | A KPI might refer to a given address of an area. |
| **KPI-14:** | A KPI might refer to an area denoted by textual geographical information such as district, borough, or other specification of the KPI coverage. |
| **KPI-15:** | A KPI can have an indicator unit. |
| **KPI-16:** | A KPI can be a measure of demography, characteristics, activity, or performance. |
| **Graphical representation** | |
|  | |
| **Use Case** | |
| In a city, it could be interesting to calculate a KPI that measures the access to vehicle sharing solutions for city travel. This KPI could be defined as the number of vehicles available for sharing per 10,000 inhabitants. This KPI could be calculated by the city council with a monthly periodicity. This KPI could refer to the principal areas of the city, for example a specific district or neighborhood. | |
| **Note** | The KPI-1 requirement has not been included in the diagram because it could be used as part of the ontology documentation. For example, this requirement has been used in the description slot of this pattern. |

The *Agent* concept, the same as explained in the Event ODP, has been included in order to represent for whom the performance is measured. In addition, this concept has been extended by means of the *Organization* in order to represent a public administration or a private entity that could calculate the KPI. Finally, the concept *EntityList* represents a list of entities whose data are aggregated. It could be represented by means of the *StructureValue* concept from the schema model, as is explained in the FIWARE data model specification.

### 4.7. Public Service ODP

The *Public Service* domain is represented in the *gci*, *km4c*, *vital*, *smart-city*, and *sctc* ontologies. In order to extract the requirements for this domain, the Core Public Service Vocabulary (https://joinup.ec.europa.eu/release/core-public-service-vocabulary/101), provided by the joinup European initiative (https://joinup.ec.europa.eu), has been taken into account. In Table 12, the proposed pattern for this domain is shown. The *Agent* concept, the same as explained in the previous subsections, has been used in order to model two requirements. First, it represents the public administration or private entity who provides a service. Second, it represents those users of the service. In addition, the *Language* concept has been included to model the different languages in which the *PublicService* is available. Moreover, the *Regulation* concept has been included to model the regulations by which the service is governed. Finally, the *Facility* concept has been modeled in order to represent those installations involved in the provision of the service.

A similar pattern definition for the public service is available in ISO/IEC 30182:2017, where a *Service* concept is used to represent the capacity to carry out one or more methods.

### 4.8. Proposed ODPs' Integrated View

After defining the domain-dependent patterns, a view of the model resulting from the integration of these patterns is presented in Figure 3. This model provides a minimum subset of the core elements that a smart city should manage in order to represent all domains and requirements specified by the patterns. According to the specific requirements of each project, these patterns may be extended in order to represent other specific classes for each domain. Due to page limitations, some subclasses from *AdministrativeArea*, *CityObject*, and *TemporalEntity* have been omitted from this view.

Even though ontologies like *km4c* and *smart-city* had covered almost all the domains shown in Table 5, they are not sufficient. Both follow a monolithic approach and do not use well-known and accepted ontologies, which makes them hard to reuse and to improve interoperability. As shown in Table 4, the *smart-city* ontology does not reuse or reference any ontology, and the *km4c* ontology is not using the *oldssn* or *saref* ontologies, which are well-known ontologies; instead of this, an ad hoc representation is used. In this context, the ODP approach allows us to design small and general patterns for each domain and then link them together in order to provide a flexible, light, and modular model, as shown graphically in Figure 3.

**Table 12.** Public Service ODP.

| Name: | Public Service |
| --- | --- |

| Description |
| --- |
| This pattern provides a model for those services supplied by the government either directly (through the public sector) or by financing the provision of services. |

| Requirements | |
| --- | --- |
| **SER-1:** | Public service is a service that is provided by the government either directly (through the public sector) or by financing the provision of services. |
| **SER-2:** | A public service can have a name. |
| **SER-3:** | A public service can have a description. |
| **SER-4:** | A public service can be available in different languages. |
| **SER-5:** | A public service can be regulated by a set of rules. |
| **SER-6:** | A public service can be related to another. |
| **SER-7:** | A public administration can provide a public service. |
| **SER-8:** | An organization can deliver a public service. |
| **SER-9:** | A person, organization, or group can use a public service. |
| **SER-10:** | A public service can be available within an area. |
| **SER-11:** | Different types of facilities can be: sport facilities, educational facilities (nursery schools, schools, colleges, universities, etc.), cultural facilities (museum, theater, libraries, etc.), facilities for the elderly, health-related facilities, leisure time facilities, restaurants, cemeteries, markets, malls, environment-related facilities, information points, animal- and plant-related facilities, social service facilities, WiFi points, parks, police offices, co-working centers, business incubators, accommodation, or mail offices. |

| Graphical representation |
| --- |
| 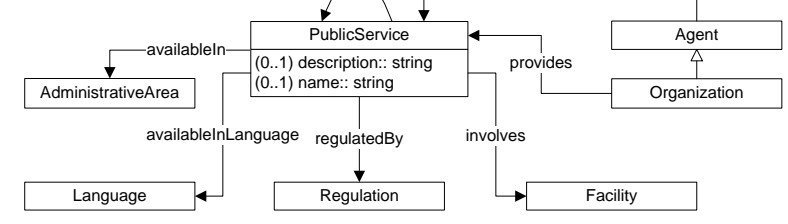 |

| Use Case |
| --- |
| Air quality control is a public service usually provided by a city council. This service allows detecting whether the air reaches a certain level of pollution within an area. When it occurs, the city council may implement a protocol in order to restrict the vehicle flow in the affected areas until the level of pollution decreases. In addition, citizens can access real-time information about air quality by means of a mobile application that may alert them when a pollution episode occurs. This kind of alert would help citizens avoid transit through the polluted place, and the mobile application may suggest an alternative route in order to redirect them. |

| Note | The SER-1 requirement has not been included in the diagram because it could be used as part of the ontology documentation. For example, this requirement has been used in the description slot of this pattern. The SER-11 requirement could be modeled in two ways. On the one hand, the *Facility* concept could be extended in order to include subclasses that represent the different types of facilities. On the other hand, *Facility* could be replaced by a specific concept from a controlled vocabulary of types of facilities. In addition, in order to represent the SER-4 requirement, the *Language* concept could be represented by means of the same named class from the Lexvo ontology (http://lexvo.org/ontology) and more specifically by a Lexvo.org language identifier (http://www.lexvo.org). Furthermore, this requirement could be represented by a link to a controlled vocabulary for language definitions, for example the European Publications Office's Languages Named Authority List (NAL http://publications.europa.eu/mdr/authority/language/index.html). |
| --- | --- |

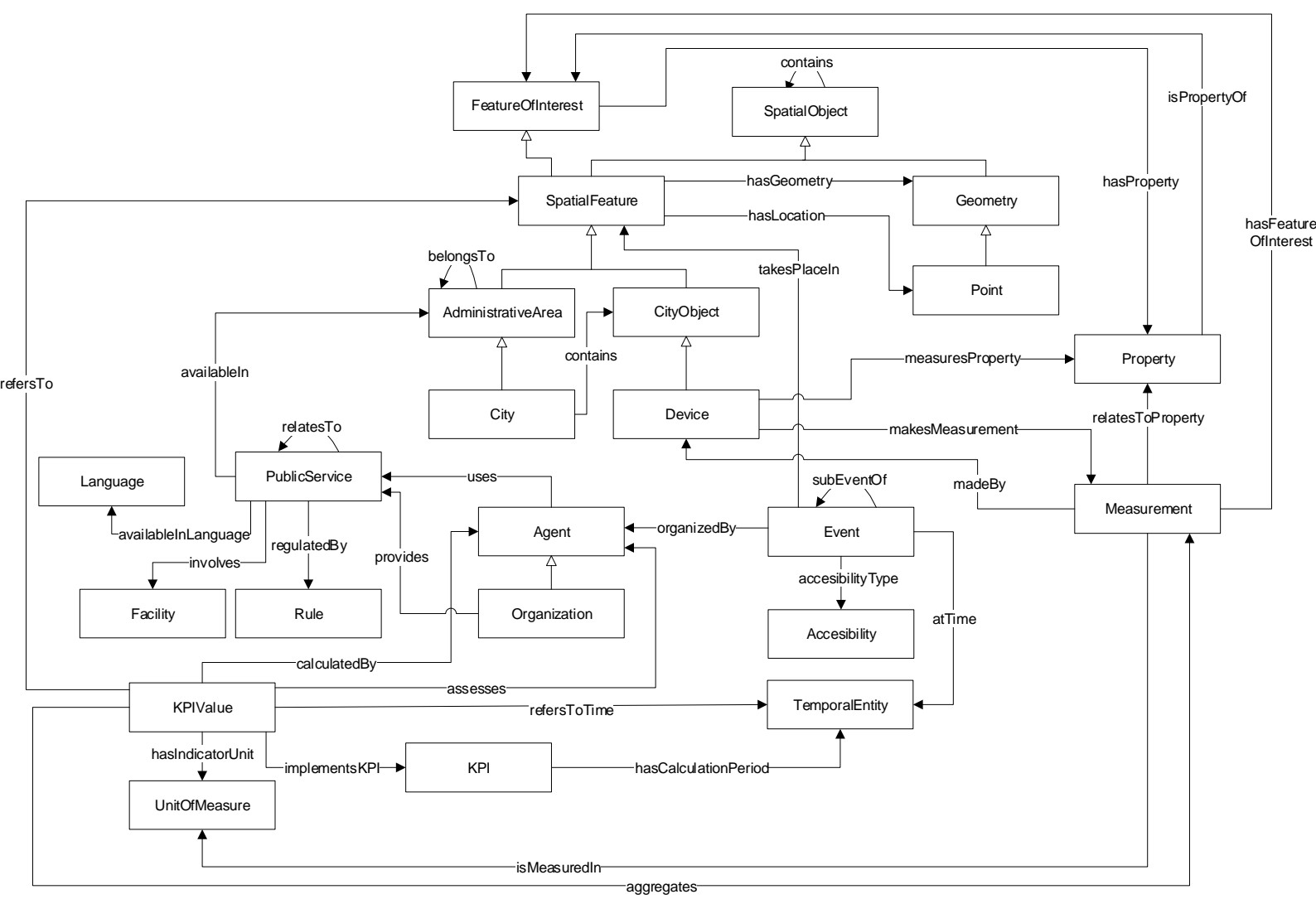

**Figure 3.** Smart city ODP overview.

## 5. Conclusions and Future Work

In this paper, a systematic literature review of smart city ontologies has been presented. The review reveals some gaps in the ontologies available for this field. First, there are smart city ontologies that are not available, a fact that completely hinders their reuse. Second, each ontology provides its custom model adapted to its specific needs, which makes it difficult to increase the interoperability among models, beyond common points, such as the modeling of time and units of measurement, where standards and well-known ontologies are usually reused. Third, some ontologies do not provide a clear documentation or the ontology requirements specification from which they are built.

This review aimed to answer the research question defined in Section 2.1 (What are the common ontology design patterns used in smart city ontologies?). As a result, a set of ontology design patterns has been proposed. These patterns materialize the domains and requirements that a smart city should represent. These seven patterns have been designed with detailed graphical conceptualizations, use cases, and suggestions about how they may be implemented, including links to the ontologies that can fit their requirements. In addition, these patterns are inspired and therefore aligned with the inspected ontologies, shown in Table 3, as well as the ontologies commonly reused by them, shown in Table 4. As a result, a core model for smart cities has been provided in order to improve the interoperability and ease the development of these kinds of ontologies.

An important aspect of the proposed ODP is the alignment with existing standards and well-known models for smart city data. Due to the fact that the proposed ODPs are based on the ontological requirements that have been obtained from standards such as "ISO 37120" and the "AENOR norm UNE 178301:2015", and broadly-used models like FIWARE, the ISA2 ontologies, etc., it is expected to achieve broader interoperability than the models designed for a particular use case.

As future work, the proposed patterns may be encoded in an ontology implementation language such as OWL (https://www.w3.org/OWL), and it may be submitted to the Ontology Design Pattern catalog. In addition, these patterns are being implemented in an ontology under development for smart cities. Moreover, it may be interesting to reach an agreement with other patterns from domains not covered in this work. Finally, a set of guidelines and examples may be provided in order to explain how to extend these patterns or how to use them with data.

**Author Contributions:** Conceptualization, M.P.-V., O.C., P.E.-A., and R.G.-C.; methodology, M.P.-V., P.E.-A., and R.G.-C.; validation, M.P.-V., and R.G.-C.; investigation, P.E.-A.; writing, original draft preparation, M.P.-V. and P.E.-A.; writing, review and editing, M.P.-V., O.C., P.E.-A., and R.G.-C.; supervision, M.P.-V. and R.G.-C.

**Funding:** This work was partially supported by a Predoctoral grant from the I+D+iprogram of the Universidad Politécnica de Madrid, ETSISpecialist Task Force 534, and DATOS4.0: RETOS Y SOLUCIONES—UPMSpanish national project (TIN2016-78011-C4-4-R).

**Acknowledgments:** The authors would like to thank the members of the ETSI Specialist Task Force 534 and of the ETSI SmartM2M Technical Committee for feedback on this work.

**Conflicts of Interest:** The authors declare no conflict of interest.

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
