# Peer review of "Ontological Representation of Smart City Data: From Devices to Cities"

_applsci, doi:10.3390/app9010032_

Reviewer 1 Report

Summary:

A survey on smart city ontologies enriches the existing survey with additional ontologies. 15 ontologies have been selected. A list of Ontology Design Patterns (ODPs) for smart cities is proposed to enrich the ODPs catalog. Several smart city ODPs have been suggested (Topology ODP, Administrative ODP, City Object ODP, Measurement ODP, Event ODP, Key Performance Indicators ODPs, Public Service ODP, proposed ODPs integrated view).

The ontology analysis focused on:

     ontology metadata (name, prefix, URI, domains, methodology, standard, group/project)

2.     the content of the ontologies

Strengths:

-        The Systematic Literature Survey (SLS) methodology followed for the literature survey [Kitchenham et al. 2007] to address this question: Which are the most common ontology design patterns used in smart cities ontologies?

-        Figure 2 is excellent

Weaknesses:

-        Table 1, Only semantic web journal and ontology catalogs are considered -> what about considering publication repositories: IEEE, MDPI, ACM, Springer, etc.,

-        Same for smart city, iot journals since they have published work about ontologies, etc. you will find much more literature about “smart city” within IoT journals, smart cities journals, etc. rather than within the semantic web community.

-        How do you model your suggested smart city ODPs? More explanations needed

-        How do you validate/evaluate your suggested smart city ODPs? More explanations needed. Do smart city experts validate them?

-        Future work explain that the ODP will be implemented

-        ODPs referenced on the ODP catalog?

Other Comments:

-        Why do you decide to select ontologies between 2012-2018?

-        Check this http://ontologydesignpatterns.org/wiki/Community:Internet_of_Things

o   Only one referenced http://ontologydesignpatterns.org/wiki/Submissions:Actuation-Actuator-Effect

-        Bullet list of metadata-oriented characterization

o   Some criteria are not explained in table 1? Is it on purpose.

o   Order in the same way the bullet list and the columns in table 1?

-        How did you build table 4? Manually or automatically?

o   If automatically provide a code example

-        Section check iot-lite ontology they mention the coverage concept (polygon, rectangle, point, circle) https://www.w3.org/Submission/2015/SUBM-iot-lite-20151126/

Literature survey suggestion:

-        Concept Extraction from Web of Things Knowledge Bases [Noura et al. October 2018] International Conference WWW/Internet.

Minor comments (e.g., typos):

-        Section Introduction -> Ontology Design Patterns(ODP), space missing

-        Section 2 Research methodology -> reverse engineering[18]

-        Redundancy footnote 1 and 2

-        Table 3 typo lecue [43] space missing before ref, same for Piui

-        Lefrancois et al. or single author?

-        Section 4 “no ontology covers all the core domains for smart cities”, find such ontologies would be challenging, however integrating them, aligning would be more feasible to get the entire knowledge to describe smart cities

-        Footnote should be before the dot?

Author Response

Dear reviewer,

In the attached file you will find our responses.

Kind regards,

Paola

Reviewer 2 Report

In this paper, authors propose a survey of existing ontologies for smart cities. A set of ontology design pattern have been identified and carefully described. The overall paper proposes a really good and interesting analyis. The research methodology has been carefully explained. While I totally agree with the choice of the ontology index and the ontology catalogs (and I appreciate the distinction), I would have preferred a wider set of journals. For example authors could consider also journals where the main focus is on the Internet of Things, due to the nature of the smart city domain.

From the point of view of the content of the paper, the Introduction should contain more information, for example related to the context of smart cities: why are smart cities so important? Are there any important investments in this area? For example the EU is financing a number of research projects involving smart cities. It would be interesting for the reader to know something more about this facts, to better appreciate the importance of the research. 

Another remark is that I really miss a detailed description of the domains listed in Table 5. This should absolutely be fixed. The reader can now something more going on reading the text, but I think that it is worth having a detailed description of the domains in the beginning of Section 4.

A technical note regarding the text: you should not define acronyms more than once (e.g., ODP, EC and IC) and you should not define acronyms that will not be used in the following (e.g., SLR, RQ). 

The quality of English is very good, but a few errors should be fixed. For example, a few remarks related to the conclusions are:

- "hinders its reuse" -> should be "their"

- "each ontology provides their" -> should be "its"

- "some ontologies does not provide" -> should be "do"

To conclude, the overall paper looks really good. It reports an interesting research work, and the research activities are described very well. A paper I have been happy to read.

Author Response

Dear reviewer,

In the attached file you will find our responses.

Kind regards,

Paola

Round  2

Reviewer 1 Report

“In most cases, authors do not include URL in their studies, or if they do it some ontologies are not online available” -> what would be the solution: send an email to the authors to encourage and educate people in publishing the ontologies, new tools to ease this task?

You might be interested in the smart city ontology dataset (a dump of ontologies for smart city can be downloaded) also a Google sheet table with smart city ontology URLs:

 http://lov4iot.appspot.com/?p=queryCityOntologiesWS

Comment “what do you decided to select ontologies between 2012-2018” -> I believe it would be nice to have the answer within the paper.

In general all answers/comments must be found in the paper.